# Fast Formation of Hydrate Induced by Micro-Nano Bubbles: A Review of Current Status

Zhiyong Jing [1], Yaxin Lin [2,*], Chuanxiao Cheng [2,*], Xiaonan Li [2], Jianxiu Liu [2], Tingxiang Jin [2,*], Wenfeng Hu [2], Yaoli Ma [2], Jiayi Zhao [2] and Shijie Wang [2]

[1]  College of Computer and Communication Engineering, Zhengzhou University of Light Industry, Zhengzhou 450002, China
[2]  School of Energy and Power Engineering, Zhengzhou University of Light Industry, Zhengzhou 450002, China
*  Correspondence: linyaxin516@163.com (Y.L.); cxcheng@zzuli.edu.cn (C.C.); txjin@126.com (T.J.)

**Abstract:** Hydrate-based technologies have excellent application potential in gas separation, gas storage, transportation, and seawater desalination, etc. However, the long induction time and the slow formation rate are critical factors affecting the application of hydrate-based technologies. Micro-nano bubbles (MNBs) can dramatically increase the formation rate of hydrates owing to their advantages of providing more nucleation sites, enhancing mass transfer, and increasing the gas–liquid interface and gas solubility. Initially, the review examines key performance MNBs on hydrate formation and dissociation processes. Specifically, a qualitative and quantitative assembly of the formation and residence characteristics of MNBs during hydrate dissociation is conducted. A review of the MNB characterization techniques to identify bubble size, rising velocity, and bubble stability is also included. Moreover, the advantages of MNBs in reinforcing hydrate formation and their internal relationship with the memory effect are summarized. Finally, combining with the current MNBs to reinforce hydrate formation technology, a new technology of gas hydrate formation by MNBs combined with ultrasound is proposed. It is anticipated that the use of MNBs could be a promising sustainable and low-cost hydrate-based technology.

**Keywords:** micro-nano bubbles; memory effect; hydrate formation; ultrasound

## 1. Introduction

Gas hydrates are ice-like solid crystalline compounds consisting of water and gas molecules, combined by hydrogen bonding to form a cage structure (the host) and gas molecules (the guest) occupying the cages to form a stable structure. It has been proved that the formation and dissociation process of a hydrate is simple; there is almost no mass loss, no environmental protection problem, and the gas storage capacity is large [1,2]. Therefore, hydrate-based technologies have shown great application potential in cold storage [3–6], gas separation [7,8], storage and transport of gases [9,10], and seawater desalination [11,12], etc. However, the characteristics of poor gas–liquid interface, the randomness of nucleation, the low rate of nucleation, and crystal growth are the key problems that restrict the fast formation of hydrate and its technology application to energy storage. Studying different reinforcement methods to achieve rapid hydrate formation is significant for promoting hydrate commercialization and industrial application.

At present, the most commonly used methods to strengthen hydrate formation are "mechanical methods [13–18]" and "additives [19–21]". The mechanical methods (e.g., stirring, spraying, and gas bubbling) introduce external energy, which significantly in-creases energy consumption, while hydrate formation can hinder the stirring. In addition, mechanical stirring is complex and challenging to apply on a large scale at high pressure and low temperature. Additives (including sodium dodecyl sulfate (SDS) [22,23], tetrabutylammonium bromide (TBAB) [24,25], tetrahydrofuran (THF) [26,27], cyclopentane (CP) [28–30], etc.) introduce impurities and disrupt the hydrate reaction system, which is

not ideal for reinforcement effects. Therefore, finding an efficient and economical method to strengthen hydrate formation without disrupting the reaction system is necessary.

Micro-nano bubbles (MNBs) are an effective method of enhancing hydrate formation without disrupting the reaction system. Compared to ordinary bubbles, MNBs have a small size (less than 1 μm in size), long residence time, high mass transfer efficiency, and high gas solubility. MNBs can provide more nucleation sites at the gas–liquid interface and reduce the conditions for hydrate formation. MNBs lead to new directions for enhanced rapid hydrate formation, different from other promotors, and are considered as effective additives to promote hydrate formation due to higher specific surface area and better heat transfer enhancement characteristics, etc. Zhang et al. [31] found that methane hydrate preferentially nucleates at the interface of MNBs. Moreover, in a hydrate dissociation solution, the MNBs generated by dissociation do not break up and disappear in a short time. They can be present throughout the formation of the hydrate dissociation solution, increasing the gas concentration in the solution and playing a pivotal role in complementing the guest molecules during hydrate formation [32]. Uchida [33] believes that MNBs can significantly increase the concentration of gases in solution to a saturated or even supersaturated state, thereby facilitating hydrate formation. High gas concentration zones generated in hydrate dissociation solutions have an important influence on the memory effect. Uchida [34] found that MNBs were present in the vicinity of hydrates for some time, and that the dissociated MNBs generated as guest molecules provided an abundant concentration of gas for hydrate formation. In addition, MNB technology has a wide range of applications in natural gas exploration, gas transportation, wastewater treatment, juice concentration, etc. Sholihah et al. [35] summarized the decomposition, formation, and recovery of hydrates from numerical simulations. Chen et al. [36] controlled the amount of MNBs so that the hydrate formed during gas transport would not cause blockages in the pipeline. Uchida et al. [34] found during natural gas exploration that MNBs can remain in solution for a long time, providing favorable conditions (memory effect) for hydrate reformation. Thus, MNB technology has applications in many fields, impacting hydrate formation, dissociation, and memory effects.

Although there is literature on the use of MNBs in hydrate solution and the association of MNBs with memory effects, there have been limited attempts to conduct a comprehensive review and organize the applications of MNBs in the field of reinforcing hydrate formation. Hence, this review organizes the influence of MNBs on the processes of hydrate dissociation and formation. Briefly, we summarize the features and advantages of MNB applications in dissociation and formation. This review has been structured to include many aspects of MNB characterization, including direct and indirect determinations of type, size, size distribution, rising speed, and bubble stability of MNBs to provide a comprehensive understanding of their properties and how they vary with bubble size. Subsequently, this paper collates the advantages and characteristics of MNBs in hydrate formation. Finally, based on the cavitation effect properties of ultrasound, an innovative experimental method to promote hydrate formation with MNBs combined with ultrasound is proposed to optimize hydrate nucleation and growth to further enhance the efficiency in hydrate-based technologies.

## 2. Generation, Residence Period, and Promotion of MNBs in Hydrate Dissociation Solution

### 2.1. Generation of MNBs during Hydrate Dissociation

It is found that guest molecules diffuse and aggregate to form MNBs in the hydrate dissociation solution during the hydrate dissociation process. Researchers have analyzed the hydrate dissociation process to generate MNBs through molecular dynamics simulations. Many techniques have been used to demonstrate the presence of MNBs in solution, with the replica SEM method being the most common method for direct observation of MNBs [37]. Using this technique, researchers have observed the presence of MNBs in the hydrate dissociation solution. Uchida et al. [34] used transmission electron microscopy (TEM) to

detect freeze-cracked samples of methane hydrate dissociation solution. It was found that MNBs still exist in the solution after the dissociation of the gas hydrates. The results show that many spherical objects are generated in the dissociation solution of methane hydrate. The methane was detected by Raman spectroscopy, which proved that the spherical objects were methane MNBs. The diameter of the MNBs ranged from 20 nm to 2000 nm, a peak number of 300–400 nm, and number concentrations of $2 \times 10^9$/mL in the sample. Similar experimental results were obtained by Takeshi [38] and Uchida [39]. This review uses the term MNBs to refer to bubbles with a diameter of 1 μm or less.

Size distribution, rising velocity, and bubble stability of MNBs affect hydrate dissociation processes. Chen [36] found that the presence of methane MNBs accelerates hydrate dissociation and that the size of the MNBs decreases. Madden [40] found that in methane hydrate dissociation solution, MNBs will continue to rise and accumulate at the gas–liquid interface. By simulating the dissociation of deep-sea hydrates, Wang [41] found that MNBs can expand and contract laterally in the dissociation solution. In addition, it has been found that the size of the MNBs generated by the dissociation solution of different hydrates is also different, as shown in Figure 1. The model is shown in Figure 1a,b of the ability of methane and nitrogen MNBs to grow laterally in the dissociation solution, whereas that displayed in Figure 1c,d presents a wide range of THF and NaCl MNBs of different sizes.

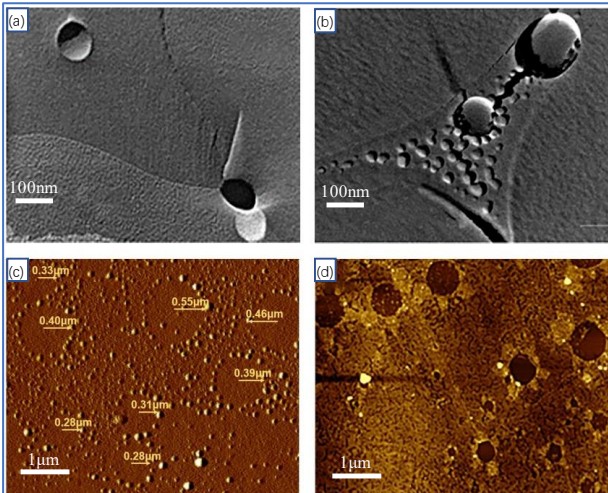

**Figure 1.** Transmission electron microscope image of MNBs in different hydrate dissociation solution: (**a**) Methane MNBs from freeze-fractured samples of hydrate dissociation solution [42], (**b**) Nitrogen MNB generation by freezing separation in frozen liquid nitrogen [43], (**c**) THF solution generates MNBs of various sizes at room temperature [44], (**d**) generating multiple sizes of MNBs in 0.01 M concentration NaCl solution [45].

In recent years, MNBs generated during hydrate dissociation also have attracted wide attention in the scaling of molecular dynamics (MD) simulations. Bagherzadeh [46] simulated the evolution of methane molecules during the dissociation of methane hydrate through MD simulations. The hydrate dissociated layer by layer, with the solution gradually increasing in methane molecules and reaching supersaturation, resulting in the generation of MNBs. During the simulation, MNBs were formed in the dissociation solution by aggregation of methane molecules. The rest diffused into the gas phase to increase the gas phase pressure and remained in the aqueous phase to maintain aqueous phase saturation. Yagasaki [47] simulated the effect of MNBs on the dissociation of methane hydrate. It was found that as the methane hydrate dissociates, the methane molecules gradually diffuse into the solution, and as the rate of diffusion of methane molecules in the solution is less than the rate of hydrate dissociation, this results in the solution being oversaturated with methane gas molecules. The generation of MNBs in the dissociation solution is shown in Figure 2. The hydrate as expected first dissociates at the outermost layer, and the rate of hydrate dissociation is greater than the diffusion rate of gas molecules in the aqueous

phase. Therefore, the gas molecules accumulate at the hydrate–liquid interface, some gas molecules will enter the gas phase or remain in the aqueous phase through diffusion, and the gas molecules accumulate at the hydrate–water interface reaching supersaturation to form MNBs.

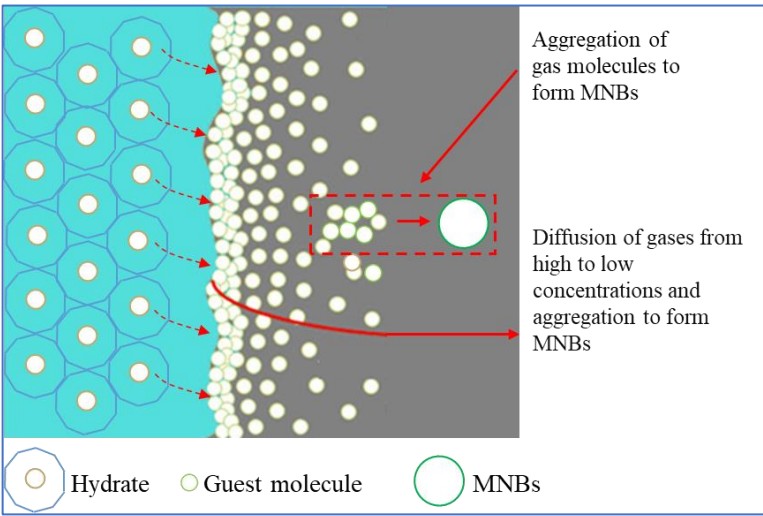

**Figure 2.** Schematic diagram of MNB generation during hydrate dissociation [34].

In summary, MNBs are generated and aggregated during hydrate dissociation, and yet the presence of MNBs requires suitable temperature and concentration conditions. In addition, there are various changes from the generation of MNBs to the secondary hydrate formation, with bubbles rising by buoyancy, becoming smaller by their own pressure, and disappearing when impeded. The MNB control at the most appropriate concentration and size is a key problem for promoting hydrate formation.

### 2.2. Residence Period of MNBs in Hydrate Dissociation Solution

In hydrate dissociation solution, the lower rise rate and lower buoyancy increase the stability of the MNBs, allowing them to persist for long periods of time. This is the key to the memory effect of secondary hydrate generation. According to the Stokes equations, the rising rate of MNBs formed by hydrate dissociation is very slow (e.g., bubble diameter 300 nm, rise speed $5.4 \times 10^{-8}$ m/s) and has a residence period of several days. Liu [48] observed the residence period of nitrogen MNBs in solution through a nanoparticle tracker and showed that the residence time of nitrogen MNBs in solution exceeds 7 days. The size, number, and density of the bubbles decrease with time due to the diffusion of the gas. Uchida [34] examined the size and residence period of MNBs generated by methane hydrate dissociation. During the residence period, it was found that the number of MNBs less than 500 nm in diameter gradually increased at the beginning and reached a maximum of 3 h after dissociation, with a peak at 300–400 nm. The increase of small bubbles is due to the contraction of large bubbles. The number of MNBs decrease with time as the methane molecules within the bubbles diffused after the completion of the dissociation, while the presence of MNBs could still be detected 4 days after the hydrate dissociation. Uchida [39] tested the dissociation solution of ethane hydrate. The result shows that the MNB size distribution varied with time. After 3 h of dissociation, the number of medium-sized MNBs decreased, and the number of both larger and smaller MNBs increased, with many MNBs still being detected in the solution after one week. Through the Uchida [34] experiments, it was found that the difference in the period of residence and size variation of MNBs generated by hydrate dissociation is related to the type of gas [49]. The dissociation of hydrate generates MNBs with a significant residence period, during which bubble size changes due to bubble shrinkage and diffusion of gas molecules, while the type of gas also affects the bubble size and residence period.

*2.3. Speeding up of Hydrate Dissociation by MNBs*

The generation of MNBs can increase the rate of heat transfer due to irregular Brownian motion and the accelerated diffusion of gases from high to low concentrations in hydrate dissociation solution. Researchers considered that the movement and rupture of MNBs in the aqueous phase could significantly increase the system's heat and mass transfer processes [50]. Yagasaki [47] found that if the thermal disturbance provided by the methane hydrate dissociation solution is insufficient to cross the energy barrier formed by the MNBs, the gas concentration in the solution is supersaturated, and the methane hydrate exists as a sub-stable superheated object. However, Uddin [51] believed that the rate of hydrate dissociation depends on the stirring rate. Localized temperature gradients and bubble build-up on the dissociating surface of hydrates significantly affect heat transfer and reduce the dissociation rate. Stirring removes the formed gas bubbles breaking the temperature gradient and causing hydrate dissociation. Ripmeester et al. [52] simulated methane dissociation and found that the rate of hydrate dissociation was influenced by the heat and mass transfer of methane molecules into the aqueous phase. The dissociation of hydrate leads to the accumulation of many methane molecules near the solid–liquid interface to form methane MNBs, which affects the mass transfer rate. It can be seen that there are different views on the effect of MNB generation on hydrate dissociation. On the one hand, the generation of MNBs absorbs gas molecules in solution to promote hydrate dissociation. On the other hand, the accumulation of MNBs at the hydrate–water interface can impede heat transfer, thus decreasing the rate of hydrate dissociation, while external disturbances need to be added to improve heat transfer to accelerate hydrate dissociation.

In summary, hydrate dissociation proceeds layer by layer, and MNBs are generated as the gas molecules accumulate. The effect of hydrate dissociation on the system's heat and mass transfer characteristics needs to be further investigated, and the combination of external disturbances would be an effective method of improving the heat transfer impeded by the accumulation of MNBs.

## 3. Rapid Hydrate Formation Induced by MNBs

*3.1. Promotion Effects of MNBs on Hydrate Formation*

The generation of MNBs by gas bubbling is one of the methods of promoting hydrate formation. Current studies on bubble-enhanced hydrate formation focus on the effect of MNBs on the induction time of hydrate formation and water conversion into hydrate. Bahadur [53,54] used the voltage method to generate (hydrogen) bubbles to promote the nucleation of tetrahydrofuran (THF) hydrate, and it was found that the control group without bubbles did not form hydrates after 15 h, while the experimental group with bubbles all nucleated within 7 min. The hydrogen bubbles generated at the electrode cathode provide many nucleation sites, while the growth of the bubbles causes convection and pressure fluctuations in the liquid, which provide activation energy for nucleation and promote hydrate nucleation. Chen et al. [36] found that using MNBs would improve the water conversion into hydrate and reduce the adverse effects of the hydrate shell, which is mechanically stable and can hinder subsequent hydrate formation. Lekse [55] studied the effect of bubble size and number on the water conversion into hydrate during hydrate growth. The results show that the smaller the bubble size, the more bubbles, and the higher is the water conversion into hydrate. In addition, Research has demonstrated that different types, sizes, and densities of MNBs can enhance hydrate formation, as shown in Table 1. Different guest molecules generate MNBs of significantly different sizes, with significant differences in number density. Among them, $CO_2$ and $H_2$ have the smallest size and are more conducive to the stable fugacity of MNBs over a long time while promoting secondary hydrate formation. MNBs have an optimal number density range: when the number density is low, MNBs do not accumulate well and adsorb hydrate nucleation and growth; when the number density is high, a large number of MNBs accumulates together to hinder the heat and mass transfer of hydrate, which is not conducive to the growth of hydrate. The same guest molecules also produce different MNBs. Furthermore, the MNBs

of $CO_2$, $C_2H_6$, $H_2$, and Xe can significantly reduce the induction time to 60 min at the right experimental temperature.

**Table 1.** Different MNBs to promote hydrate formation.

| Source | Size Distribution (nm) | Number Density ($10^8$/mL) | Induction Time (min) | Store Temperature (°C) | Reference |
|---|---|---|---|---|---|
| Pure water | 0 | 0 | 2.5–67 | 20 | [38] |
| Air | 140–350 | 2–9 | - | 25 | [56] |
| Air | 100–120 | 2–9 | - | 25 | [57] |
| $O_2$ | <200 | - | - | 25 | [58] |
| $CO_2$ | 80–130 | 1–3 | 11.1–37.2 | 1–6.2 | [59] |
| $CO_2$ | 514.5 | >0.06 | 3–10 | 5 | [60] |
| $CH_4$ | 220–300 | - | 14.2–38.6 | 1–4.5 | [61] |
| $CH_4$ | 80–240 | - | 10–18 | 11–84 | [62] |
| $C_2H_6$ | 200–300 | 20–50 | 0.5–6.5 | 5–20 | [39] |
| $C_3H_8$ | >300 | >0.06 | 1.8–14.1 | 0–5 | [33] |
| $C_3H_8$ | 180–290 | 1–3.16 | 14.1–38.0 | 1.5–3.5 | [63] |
| $N_2$ | 200–300 | 20–50 | 5.5–56.5 | 5–20 | [39] |
| $N_2$ | 160–350 | - | 14.5–47.2 | 1.5–15.5 | [64] |
| $H_2$ | 50–150 | 7–8 | 8.6–55.3 | 1.5–7 | [65] |
| $SF_6$ | 100–240 | 7.6–192 | - | - | [66] |
| Xe | 200–300 | 20–50 | 4.7–10.3 | 5–20 | [39] |
| Ozone | 265 | - | - | 20 | [67] |
| Ozone | 266 | - | - | 21 | [67] |
| Ozone | 267 | - | - | 22 | [67] |

It can be seen that MNBs effectively promote hydrate formation, which can significantly shorten induction time, but the hydrate shell formed around the bubble can hinder further hydrate formation and decrease the water conversion into hydrate. MNB increased concentration of bubbles helps increase the water conversion into hydrate.

*3.2. Advantages of MNBs for Reinforced Hydrate Formation*

3.2.1. Providing More Nucleation Sites

With their small size and large specific surface area, MNBs provide nucleation sites for hydrate nucleation. The use of MNBs as nucleation sites has been demonstrated by researchers from microscopic observations and crystal growth. At the microscopic scale, Zhao [68] investigated the effect of MNBs on hydrate nucleation morphology. The process of crystal morphology change was obtained by high-resolution polarized light microscopy, where hydrates were found to nucleate preferentially at the MNB interface.

Besides, the hydrophobic surface plays an essential role in promoting the formation of MNBs, and through comparative crystal growth experiments, Knott [69] found rapid nucleation in the solution of glycine-containing argon MNBs. These act as nucleation sites to promote crystal nucleation, as shown in Figure 3. In the MNB system, MNBs can provide a substantial variety of nucleation sites for hydrate, so that a substantial variety of hydrates can be attached and deposited on the surface of MNBs, greatly promoting hydrate nucleation. Fatemi et al. [70] proposed a method to increase the inhomogeneous interface by injecting MNBs to improve nucleation in a continuous microfluidic device. It was found that adding microbubbles caused the crystals to become more prominent and the induction time to shorten simultaneously. In addition, after adding MNBs, crystallization occurred at the gas–liquid interface. Although some crystallization is also found in the aqueous phase, the crystals in the aqueous phase grow and move towards the gas–liquid interface over time, and therefore the pipeline would not be blocked, indicating that the addition of MNBs changes the nucleation and growth position of crystals. SEM imaging also confirmed that the MNBs do not affect the morphology of the crystals, and the authors concluded that

the bubble interface could be used as a non-homogeneous nucleation site to promote the non-homogeneous nucleation of hydrates.

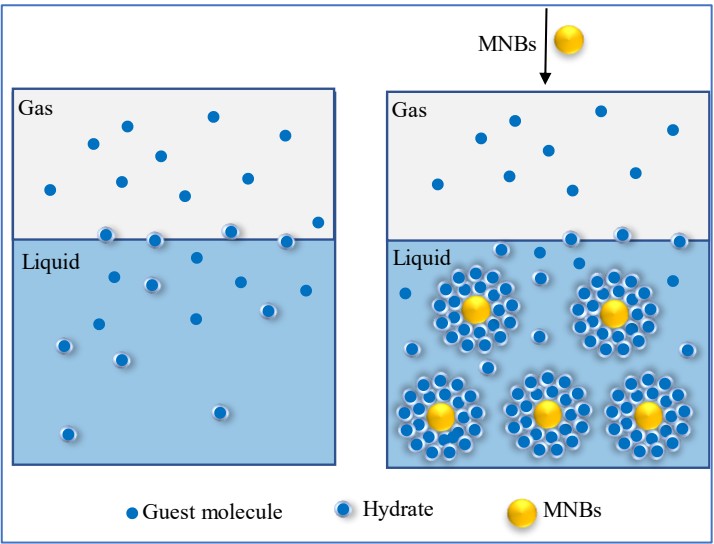

**Figure 3.** Diagram of how the addition of microbubbles changes the nucleation position of crystals [70].

In summary, it is clear that hydrates preferentially nucleate on the surface of the MNBs, while causing hydrate growth toward the bubbles. The MNBs provide nucleation sites for hydrates, promote non-homogeneous nucleation, and increase the nucleation rate while changing the nucleation location and growth direction of hydrates.

### 3.2.2. Increasing Gas Solubility

MNBs have been found to have a significant advantage in promoting hydrate formation, and researchers believe that MNBs play an essential role in increasing gas solubility [71]. Oshita [72] examined changes in MNB size and dissolved oxygen concentration in a solution by adding oxygen MNBs. The results show that the change in dissolved oxygen concentration is consistent with the change in the particle size distribution curve of the MNBs over time, indicating that the presence of the MNBs increases the dissolved oxygen concentration in the solution. In addition, Kusalik [73] simulated the nucleation of hydrogen sulfide hydrate and found that the spherical MNB structure leads to higher hydrogen sulfide solution concentrations, significantly increasing the possibility of coordination of second ortho-position (in the range of 5.5–7.5 Å), which enhances the tendency of cage formation and shortens the induction time of the gas hydrates.

### 3.2.3. Increasing the Driving Force of Hydrate Nucleation by MNBs

The self-contracting pressurized dissolution and collapse characteristics of MNBs increase the nucleation driving force to promote nucleation. It has been demonstrated that pressure fluctuation generated by collapsing MNBs promotes hydrate formation [74,75]. The smaller the bubble diameter, the higher the internal pressure and the higher is the driving force, which facilitates nucleation and further hydrate growth. Takahashi [76] used xenon MNBs to promote xenon hydrate formation, and it was found that the addition of MNBs accelerates hydrate nucleation even though the overall system temperature did not reach hydrate nucleation conditions. The hydrate formation process can be described as follows: first, the MNBs are present in the solution with self-contraction; subsequently, the internal pressure of the MNBs increases; finally, the self-contraction solution of the MNBs provides a greater drive for nucleation to approach or exceed the subcooling limitation, facilitating hydrate formation, as shown in the model diagram in Figure 4. Due to the high

internal pressure and self-contraction effect of the MNBs, they have a great potential to promote hydrate formation and shorten induction time.

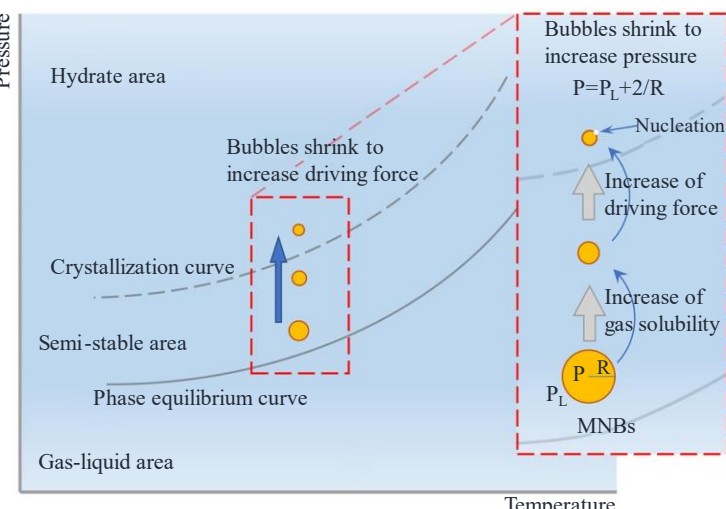

**Figure 4.** Schematic diagram of increasing the driving force of hydrate nucleation by MNBs [77].

In summary, MNBs not only provide more nucleation sites but also increase the gas–liquid interface and enhance heat and mass transfer. MNBs can act as a third interface, providing a new interface for some insoluble gases thereby greatly enhancing hydrate nucleation. However, MNBs are homogeneously distributed in solution and present for a long time until hydrate nucleation makes them one of the key issues in promoting hydrate nucleation.

### 3.3. MNBs Reinforce Hydrate Formation with Memory Effect

The memory effect refers to observations that gas hydrates form easier or faster from water obtained by the dissociation of gas hydrates or melting of ice [78,79]. The memory effect offers an effective way to form gas hydrates with better reproducibility in otherwise highly stochastic nucleation phenomena. The memory effect is known to vanish at a sufficiently high superheating temperature and/or with sufficiently long heating time when the thermodynamic equilibrium of the system is fully established. Moreover, MNBs will also disappear due to their own pressure and shrinkage properties at high superheating temperature and long-term storage.

The responsible mechanism of memory effect mainly consists of the following three hypotheses: (i) residual structure hypothesis, indicating that the hydrate-like structure remains in the decomposed solution and can provide nucleation sites for hydrate reformation; (ii) gas supersaturation hypothesis, indicating that the retained gas nanobubbles in solution after hydrate dissociation can affect the thermodynamic state of solution and the nucleation process of hydrate; (iii) impurity imprinting hypothesis, indicating that the imprinted surface of porous media driven by hydrate formation and dissociation can benefit the hydrate reformation [79]. Buchanan [80] used a combination of neutron diffraction and gas consumption to study the structure of water before and after hydrate formation, which showed no significant differences between the water structure itself and the water surrounding the methane before and after hydrate formation. Uchida [39] studied hydrate dissociation using simultaneous X-ray computed tomography, showing that the water's residual structure was not found to be the direct cause of the 'memory effect,' but which is much more associated with the enrichment of the guest molecules into MNBs in the aqueous phase. Rodger [32] believed that the memory effect of methane hydrate is caused by the high concentration of methane in the solution due to the slow diffusion of methane after dissociation. Similarly, Zhang [30] studied the dissolution properties of $CO_2$ throughout the induction time and concluded that the 'memory effect' of hydrate should not be

attributed to the residual structure. Instead, the residual gas molecules in the aqueous solution and the gas concentration are the key factors in hydrate nucleation. It was found that the memory effect could be produced by adding small amounts of dissociation solution to pure water, suggesting that the memory effect exists in the aqueous phase and that some form of the residual surface may be accelerating the secondary hydrate formation [81]. It has been suggested that the self-pressurizing solubilization properties of MNBs provide a higher saturation of the guest molecules and facilitate the memory effect.

In addition, Uchida [34] and Bagherzadeh [82] found a correlation between the memory effect and MNBs, where MNBs are present at a fixed location near the hydrate for some time, and the presence of bubbles enhances hydrate formation. The MNBs generated by hydrate dissociation act as a storage material for guest molecules in the memory effect, providing a rich concentration of guest molecules for hydrate formation and keeping them at a high local concentration during secondary hydrate formation. Uchida [39] experimented with ethane hydrate dissociation solution and ethane MNB solution, where the presence of gas in the form of MNBs allows the guest molecules to remain supersaturated in the solution. At the same time, due to the high Laplace pressure, the MNBs undergo a self-shrinking effect, becoming smaller and smaller in size and finally collapsing in the aqueous phase, further increasing the local gas concentration. In addition, heating or prolonged storage causes more gas molecules in the aqueous phase to escape to the gas phase, decreasing gas saturation in the aqueous phase and weakening the memory effect.

According to the above study, the "memory effect" of the hydrate is related to the presence of MNBs in the dissociation solution. In the hydrate dissociation, MNBs are formed and exist in the dissociation solution for a long time, which increases the concentration of guest molecules in the aqueous phase and provides a mechanism for the secondary formation of hydrate to maintain the guest molecules at the local concentration of supersaturation, thus shortening the induction time of secondary hydrate nucleation. However, how to ensure the effective time of the memory effect or the effect of the memory effect in multiple cycles? In a word, coupling the most appropriate temperature and concentration of MNBs at the effective time and cycle index of the memory effect is a key issue in reinforcing hydrate formation.

## 4. New Method for Reinforced Hydrate Formation by MNBs

Based on the above assembly of the advantages and disadvantages of MNBs in hydrate formation, a novel method of reinforcing hydrate formation by combining MNBs with ultrasound is proposed. Cheng et al. [62] found that the size of MNBs in the 80–240 nm range reduced the average nucleation induction time of hydrate by 80.9% and increased the average nucleation rate by 4.73 times. The large specific surface area of MNBs provides more nucleation sites for hydrate and enhances heat and mass transfer. Furthermore, Zhong et al. [83] found that hydrate could be rapidly formed when 100 W of ultrasound was applied to cold water for 5 s. Ultrasound can provide more energy for hydrate nucleation yet introduce more heat into the system. The combined MNBs with ultrasound avoids these difficulties, provides more energy for hydrate nucleation, and increases heat and mass transfer. This method has the following advantages in reinforcing hydrate nucleation and growth: First, under the action of ultrasound, MNBs are broken into more and smaller bubbles, which provide more nucleation sites. At the same time, the high pressure formed by bubble collapse (inside pressure of 70–150 kPa and outside pressure of 440–605 kPa [84]) provides the driving force for nucleation. Ultrasound increases the heat and mass transfer while providing energy for hydrate growth. Finally, the hydrate shell around the bubbles is broken up by ultrasound mechanics, renewing the gas-liquid interface, while the broken hydrate crystals act as secondary nucleation sites to promote hydrate growth.

The principle of ultrasound combined with MNBs for rapid hydrate formation is shown in Figure 5. Hydrate formation has disadvantages in pure water systems, such as long induction time, small gas–liquid interface, and non-uniform nucleation. Adding MNBs to the solution can significantly shorten induction time and accelerate hydrate

formation since MNBs increase the gas–liquid contact area and act as hydrate nucleation sites. At the same time, the high Laplace pressure provides a higher gas concentration for the surrounding liquid, which facilitates hydrate nucleation. Although MNBs can promote the nucleation of hydrates, they are not ideal for promoting hydrate growth. Therefore, ultrasound combined with MNBs is introduced to enhance the formation of hydrate. First, cavitation causes larger bubbles to break into smaller ones, providing more nucleation sites to promote nucleation. Second, ultrasonic vibrations increase heat and mass transfer while providing energy for hydrate growth. Finally, under the action of ultrasonic mechanics, the primary hydrate shell breaks into smaller nuclei to promote secondary nucleation and further accelerates hydrate growth.

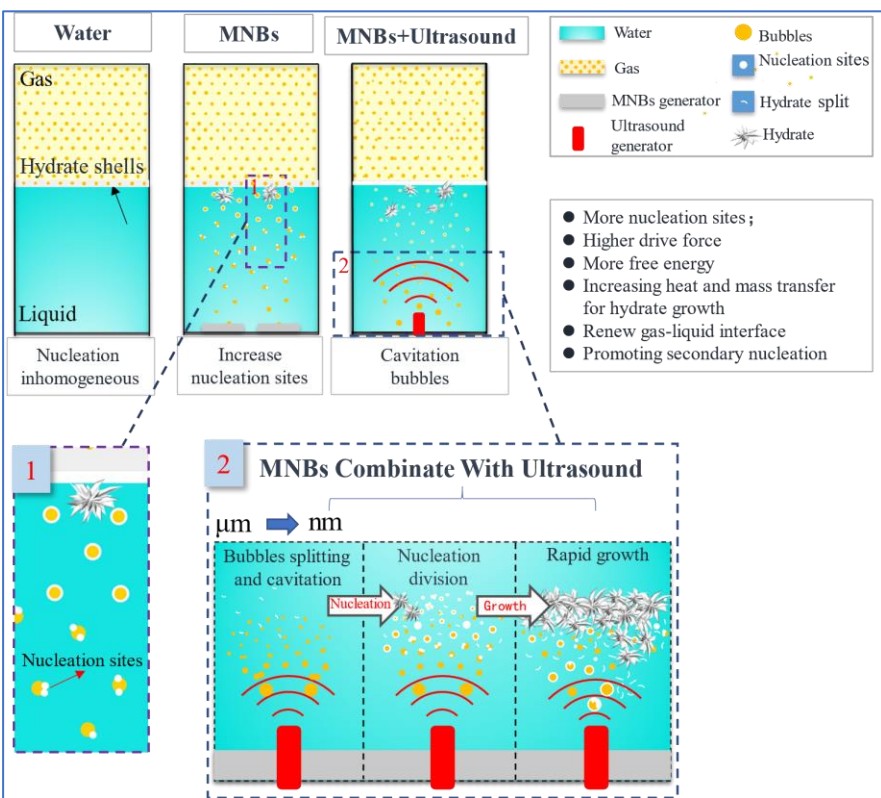

**Figure 5.** Schematic diagram of the mechanism of MNBs combined with ultrasound-reinforced hydrate formation.

Based on these principles, an experimental apparatus for MNBs combined with ultrasound to reinforce hydrate formation was designed and developed. The system contains the following five components: MNB solution generation and ultrasound section, thermostatic water circulation section, data acquisition and visualization section, and gas input and output section. In the MNB solution generation and ultrasound section, the MNB solution is generated by an MNB generator and injected into the reactor. An ultrasonic reactor section with visualization windows on both sides of the reactor facilitates observation of the hydrate formation process. The ultrasound generator is hermetically connected to the reactor and generates ultrasound at different intensities. The MNB solution generation and ultrasonic reactor section are the system's core. The thermostatic water circulation section, in which the ultrasonic reactor is placed in thermostatic water circulation, controls the temperature inside the reactor with a thermostat. In the data acquisition section, pressure and temperature transducers are used to observe the pressure and temperature changes during the experiment and record the information, while the camera unit is used to capture changes in hydrate morphology. The gas input and output sections control the inflow and outflow of gas into and out of the reactor.

The fast hydrate formation characteristics of MNBs are combined with ultrasound through data analysis and visualization. Induction time and hydrate growth rate are determined by the pressure and temperature variation with time. The most suitable conditions for hydrate nucleation and growth are explored by studying the effects of different ultrasound intensities, bubble concentrations, and bubble types. It should be noted that ultrasound has significant geothermal effects, and the appropriate ultrasound operating time should be chosen, as shown in Figure 6. Moreover, the effect of ultrasound on hydrate nucleation and growth varies depending on the hydrate morphology [50].

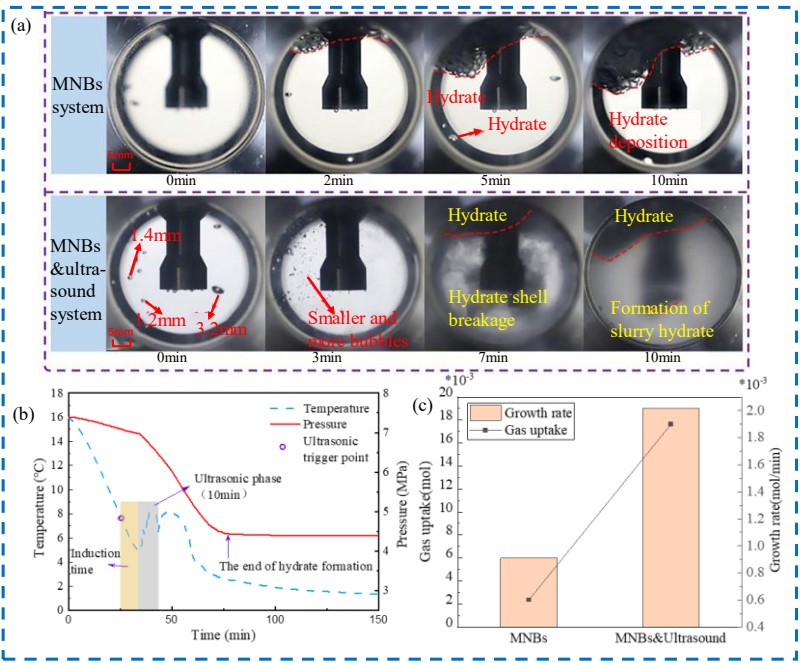

**Figure 6.** (**a**) Visualization of hydrate growth morphology by MNB system and ultrasound combined with MNB system, (**b**) ultrasound combined with MNB temperature curves, (**c**) ultrasound combined with MNBs on growth rate and gas uptake [49].

## 5. Conclusions

This paper reviews the research literature on the mechanism of the role of MNBs in hydrate formation and dissociation, describes the mechanism of the role of MNBs in reinforcing hydrate formation, discusses the effect of MNBs on hydrate dissociation and memory effect, and proposes a novel experimental idea of MNBs combined with ultrasound to reinforce hydrate formation. The main conclusions are as follows.

(1) MNBs can exist in solution for long durations, and the characteristics of their small size, large specific surface area, self-contraction, and self-collapse can provide more nucleation sites for hydrates, promote non-homogeneous nucleation of hydrates, and provide local pressure driving force for hydrate formation.

(2) MNBs can be present in solution for a long time thereby increasing the concentration of gas molecules in the liquid phase and causing it to be in a saturated/supersaturated condition, forming a local concentration difference, and shortening the induction time for secondary hydrate formation.

(3) A method and apparatus for MNBs combined with ultrasound-enhanced hydrate formation are proposed. The MNBs act more on the hydrate nucleation stage, and the introduction of ultrasound enhances hydrate nucleation and growth. The thermal effect of ultrasound can hinder hydrate formation, and attention should be paid to the effect of ultrasound action time and intensity at different stages.

**Author Contributions:** Z.J.: Conceptualization, Writing—original draft. Y.L.: Methodology, Writing—Review and editing, C.C.: Review and editing. X.L.: Data curation. J.L.: Software. T.J.:

Project administration, Visualization. W.H.: Project administration, Visualization. Y.M.: Data curation. J.Z.: Formal analysis. S.W.: Formal analysis. All authors have read and agreed to the published version of the manuscript.

**Funding:** This work was supported by the National Natural Science Foundation of China (51606173, 51606172, 51622603, and 52006024), Science and Technology Innovation Talents Support Program for colleges and universities in Henan Province (23HASTIT017), the PhD Research Funds of Zhengzhou University of Light Industry (2021BSJJ044), Henan Provincial Key Young Teachers Training Program (2020GGJS128), and Science and Technology Department of Henan Province (232102321083).

**Data Availability Statement:** Data openly available in a public repository.

**Conflicts of Interest:** The authors declare that they have no known competing financial interest or personal relationships that could have appeared to influence the work reported in this paper.

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
