# Peer review of "Fast Formation of Hydrate Induced by Micro-Nano Bubbles: A Review of Current Status"

_processes, doi:10.3390/pr11041019_

Round 1

Reviewer 1 Report

Dear Authors, kindly check for the attachment review. 

Reviewer 2 Report

1.  The paragraph between lines 378-391 can be written more clearly. First, all five elements of the experimental apparatus should be named: the MNBs solution generation section, the ultrasonic reactor section, the thermostatic water circulation section, the data acquisition section, and the gas input and output sections. Then the particular sections can be described one by one in a more detailed manner.

2.  Line 74:  velocity is to replace by speed

3.  Line 366: Laplacian to replace by Laplace.

4.  What are the magnitudes of inside and outside pressure when the ultrasound method is combined with MNBs?

5.  For what conditions is the number density specified in Table 1?

6.  The authors do count some relevant processes through Abstract and Introduction Section. Nevertheless the possible areas of practical application of MNBs in hydrate-based technology including ultrasound method can be described in a more detailed manner. That could even enhance the reader’s interest.

7.  Missing space (places highlighted in yellow). There is a need to make space between words or other parts of text in lines: 12, 30, 32, 33 ,39, 43, 44, 55, 59, 67, 201, 202, 207, 210, 220 (table 1), 230, 234, 239, 252, 260, 264, 272, 274, 285, 294, 308, 311, 315, 317, 323, 326, 332, 399, 403. Just use a space bar.

Examples:

·         Line 12:  bubbles(MNBs) to replace by bubbles (MNBs)

·         Line 30:  large[1,2] to replace by large [1, 2]

·         Line 59: Uchida[33] to replace by Uchida [33]    

8. Hyphen to remove (words highlighted in green)

 Line 34:   prob-lems to replace by problems

 Line 60: con-centration to replace by concentration

Line 212: Re-search to replace by research

9.  Other changes (highlighted in blue)

 Line 74: bubble stability to remove bubble

 Figure 4. (Bubbles) shrinks to replace by shrink, i.e., remove s

10.  The letters at Figure 6. are of different size and are placed in a way which do not facilitate readability

11.  There are some acyclic saturated hydrocarbons, including methane and ethane listed in Table 1 as the sources of MNBs. In the text (line 218) ethylene is mentioned, but is not included within the Table 1.
